# β-amyloid monomer scavenging by an anticalin protein prevents neuronal hyperactivity in mouse models of Alzheimer's Disease

Benedikt Zott [1,2,3,4] ✉, Lea Nästle[5], Christine Grienberger [1,6], Felix Unger [1,2,3], Manuel M. Knauer [1], Christian Wolf [1,2], Aylin Keskin-Dargin[1], Anna Feuerbach[5], Marc Aurel Busche [1,7], Arne Skerra [5] ✉ & Arthur Konnerth [1,4] ✉

Hyperactivity mediated by synaptotoxic β-amyloid (Aβ) oligomers is one of the earliest forms of neuronal dysfunction in Alzheimer's disease. In the search for a preventive treatment strategy, we tested the effect of scavenging Aβ peptides before Aβ plaque formation. Using in vivo two-photon calcium imaging and SF-iGluSnFR-based glutamate imaging in hippocampal slices, we demonstrate that an Aβ binding anticalin protein (Aβ-anticalin) can suppress early neuronal hyperactivity and synaptic glutamate accumulation in the APP23xPS45 mouse model of β-amyloidosis. Our results suggest that the sole targeting of Aβ monomers is sufficient for the hyperactivity-suppressing effect of the Aβ-anticalin at early disease stages. Biochemical and neurophysiological analyses indicate that the Aβ-anticalin-dependent depletion of naturally secreted Aβ monomers interrupts their aggregation to neurotoxic oligomers and, thereby, reverses early neuronal and synaptic dysfunctions. Thus, our results suggest that Aβ monomer scavenging plays a key role in the repair of neuronal function at early stages of AD.

How to halt Alzheimer's disease (AD) and the associated cognitive decline and memory loss remains one of the major challenges in the research of brain diseases. According to the β-amyloid (Aβ) hypothesis, Aβ peptides and, especially, their soluble aggregates comprising dimers and small oligomers are the most toxic agents that perturb the integrity of neuronal structure and function[1]. Aβ peptides were also shown to cause inflammation[2], tau hyperphosphorylation[3], and, ultimately, cell death[4]. Given these observations, the depletion of Aβ is expected to slow down or,

hopefully, even prevent the cognitive decline associated with AD in affected individuals.

Until relatively recently, treatment strategies aiming at the scavenging of Aβ by passive immunization with antibodies have largely failed to show a significant deceleration of cognitive decline in clinical studies, in fact raising concerns about the amyloid hypothesis[5]. Similar discouraging observations were made in Alzheimer's mouse models[6,7]. There are several possible explanations for the general failure of these approaches, most prominently an inadequate timing of the

[1]Institute of Neuroscience, Technical University of Munich, Munich, Germany. [2]Department of Neuroradiology, MRI hospital of the Technical University of Munich, Munich, Germany. [3]TUM Institute for Advanced Study, Garching, Germany. [4]Munich Cluster for Systems Neurology (SyNergy), Munich, Germany. [5]Chair of Biological Chemistry, Technical University of Munich, Freising, Germany. [6]Department of Biology and Volen National Center of Complex Systems, Brandeis University, Waltham, MA, USA. [7]UK Dementia Research Institute at UCL, University College London, London, United Kingdom. ✉e-mail: benedikt.zott@tum.de; skerra@tum.de; arthur.konnerth@tum.de

therapeutic intervention after the brain has already undergone irreparable damage[8]. However, groundbreaking results were reported for the monoclonal antibodies (mAb), foremost Lecanemab, which preferentially binds protofibrils (> 75 kDa soluble Aβ aggregates) and significantly reduced cognitive decline in patients with early AD in a phase 3 study[9].

Moreover, anti-Aβ treatment at very early stages of development had positive outcomes in mouse models of β-amyloidosis. Thus, before Aβ plaque formation, the prevention of extracellular Aβ accumulation involving the use of γ-secretase inhibitors can effectively abolish neuronal hyperactivity[10]. This is relevant because a variety of studies in mice and humans have established that neuronal hyperactivity is probably the earliest form of neuronal dysfunction in the diseased brain[10–17]. This neuronal hyperactivity can be induced directly by the application of soluble Aβ dimers, the smallest of all oligomers, and develops before Aβ plaque formation[10]. Mechanistically, Aβ-dependent neuronal hyperactivity has been linked to aberrant synaptic glutamatergic transmission[18] and impairments of inhibitory neurons[19].

To untangle the conflicting results on the effectivity of Aβ removal obtained from previous mouse studies[6,10,20], we applied here an alternative approach based on the direct intracerebral application of Aβ-binding anticalins[21]. We asked (1) whether Aβ removal can restore neuronal and synaptic functions in mouse models of AD and (2) which species of Aβ should be targeted for such an approach to be successful. Anticalins are proteins selected via phage display from a random library based on the human lipocalin protein scaffold and further engineered for optimal pharmacological properties. Anticalins exhibit very high target affinities and, in contrast to antibodies, a low immunogenic potential[22]. These properties make them potent Aβ scavengers and the ideal tool to study the acute effects of Aβ removal in vivo. Several Aβ-binding anticalins with high affinities and specificities for the monomeric Aβ peptide target have recently been described[23], and their mode of tight complex formation with the central Aβ epitope (Lys$^{P16}$ to Lys$^{P28}$)—which is common to both $Aβ_{40}$ and $Aβ_{42}$ peptide species—was elucidated by X-ray crystallography[24].

## Results

### Aβ-anticalins prevent neuronal and synaptic dysfunctions

We produced the anticalin H1GA (dubbed Aβ-anticalin) (Fig. 1A) in Escherichia coli and purified it to homogeneity as previously described (Fig. S1, materials and methods). The recombinant protein was biochemically characterized by ESI mass spectrometry and confirmation of binding activity towards the monomeric Aβ(1-40) peptide using real-time surface plasmon resonance (SPR) spectroscopy. In our experimental setup, this Aβ-anticalin solution was directly applied in vivo to the exposed hippocampal CA1 region[10,18] of 2–3-month-old APP23xPS45 mice, at a stage preceding the formation of Aβ plaques[14]. In this way, problems of brain delivery involving systemic application of this protein were circumvented. Simultaneously, we monitored changes in neuronal activity at single-cell resolution by using two-photon calcium imaging (Fig. 1 B, C).

In line with previous observations made for the same animal model under similar conditions[10], we registered a marked hyperactivity in a fraction of neurons under baseline conditions (Fig. 1D left). Local application of Aβ-anticalin to the monitored CA1 neurons rapidly suppressed the hyperactivity (Fig. 1D middle, S2A). Notably, the Aβ-anticalin effect on neuronal activity was largely reversible after a washout period of only 5–10 min (Fig. 1D right, S2B).

We have previously established that, at these early stages, neuronal dysfunction of hippocampal CA1 pyramidal neurons in mouse models of β-amyloidosis is characterized mainly by the emergence of hyperactive neurons, i.e., neurons with baseline activity levels of > 20 $Ca^{2+}$ transients/min[10,18]. Aβ-anticalin treatment reduced the number of hyperactive cells in APP23xPS45 mice to the same levels observed in wild-type mice (Fig. 1E). Moreover, a cell-by-cell analysis revealed that

Aβ-anticalin treatment reduced neuronal activity levels in all neurons (Fig. 1F, Fig. S2 C–E), but most strongly affected the group of hyperactive cells (Fig. S3). Overall, in APP23xPS45 mice treated with the Aβ-anticalin, the neuronal activity distribution was undiscernible from that of wild-type mice (Fig. 1F).

Next, we estimated the minimum pipette concentration of the Aβ-anticalin necessary to reduce neuronal activity levels. To this end, we applied decreasing concentrations of the Aβ-anticalin to the CA1 hippocampal area of non-plaque-bearing APP23 mice, a model of early AD, which, due to the lack of the PS mutation, develops plaques at a later age[25]. These mice, as the APP23xPS45 mice, had high levels of hyperactive neurons (Fig. S4A) at the age of 4–8 months. The application of the Aβ-anticalin markedly reduced neuronal activity levels in APP23 mice, also at low pipette concentrations ($IC_{50}$ 75 nM) (Fig. S4B). However, as any solution applied from the point source of a pipette tip mixes rapidly with the interstitial fluid and becomes progressively diluted with increasing distance to the tip[26], we expect the actual tissue concentration at the neuronal membranes to be much lower than in the pipette.

Control experiments demonstrated that the application of the Aβ-anticalin did not alter activity levels in 2–4 month-old wild-type mice (Fig. S5, A and B), including those of the few neurons classified as hyperactive under baseline conditions (Fig. S5C). Likewise, the application of the recombinant human lipocalin 2, the protein from which the anticalin originates and which does not bind Aβ in vitro[23], in young APP23xPS45 mice did not change the average number of $Ca^{2+}$ transients (Fig. S5D) or the number of hyperactive neurons (Fig. S5E).

Assuming that the Aβ-anticalin effect is mediated by the reduction of toxic Aβ-species in the brain, other means of Aβ removal should also be effective in preventing neuronal hyperactivity in AD mice. Thus, we next investigated the action of the monoclonal Solanezumab antibody, which binds Aβ at the same epitope as the Aβ-anticalin[24,27], has a high affinity for Aβ monomers, but not oligomers or fibrils[28] and interacts with plaques in AD mouse models[29]. The application of Solanezumab into the hippocampal CA1 region of 4–8 month-old APP23 mice decreased overall neuronal activity levels (Fig. S6, A–C), albeit less effectively than the Aβ anticalin at the same concentration (Fig. S6D). In contrast, the application of a control IgG (Fig. S6E) or of the lipocalin 2 protein (Fig. S6F) was ineffective. Second, we applied the γ-secretase inhibitor LY-411575, which blocks the biosynthesis of the Aβ peptide[10,30]. Indeed, γ-secretase inhibition markedly reduced the number of hyperactive cells and overall activity levels in 2–4 month-old APP23xPS45 mice (Fig. S7 A, C, D), but not in wild-type mice (Fig. S7 B–E).

At the synaptic level, Aβ-dependent neuronal hyperactivity is associated with the accumulation of extracellular glutamate[18,31–33]. To investigate whether the Aβ-anticalin prevents this synaptic impairment, we performed two-photon imaging of synaptically released glutamate in hippocampal slices. First, using a viral vector, we expressed the fluorescent glutamate sensor SF-iGluSnFR3[34] unilaterally in the hippocampal CA1 region of wild-type and APP23xPS45 mice in vivo (Fig. 2A, top). After 2–3 weeks of viral expression, when the animals were 2–4 months old, we performed two-photon population imaging of synaptically evoked glutamate transients in the stratum radiatum of CA1 (Fig. 2A, bottom) by electric stimulation of the Schaffer collaterals[35]. We have previously reported that Aβ-induced neuronal and synaptic dysfunctions are dependent on the level of baseline synaptic transmission[18]. Accordingly, we observed increasingly larger glutamate responses in the APP23xPS45 mice compared to the wild-type animals for higher numbers of stimuli, but no significant difference for low numbers of pulses (Fig. 2 B, C). The application of the Aβ-anticalin in APP23xPS45 mice reliably reduced the synaptically evoked glutamate transients to levels similar to those observed in wild-type animals (Fig.2 D, E). In control experiments, the application of the lipocalin 2 protein to APP23xPS45 mice (Fig. 2F, Fig. S8A and C) or of

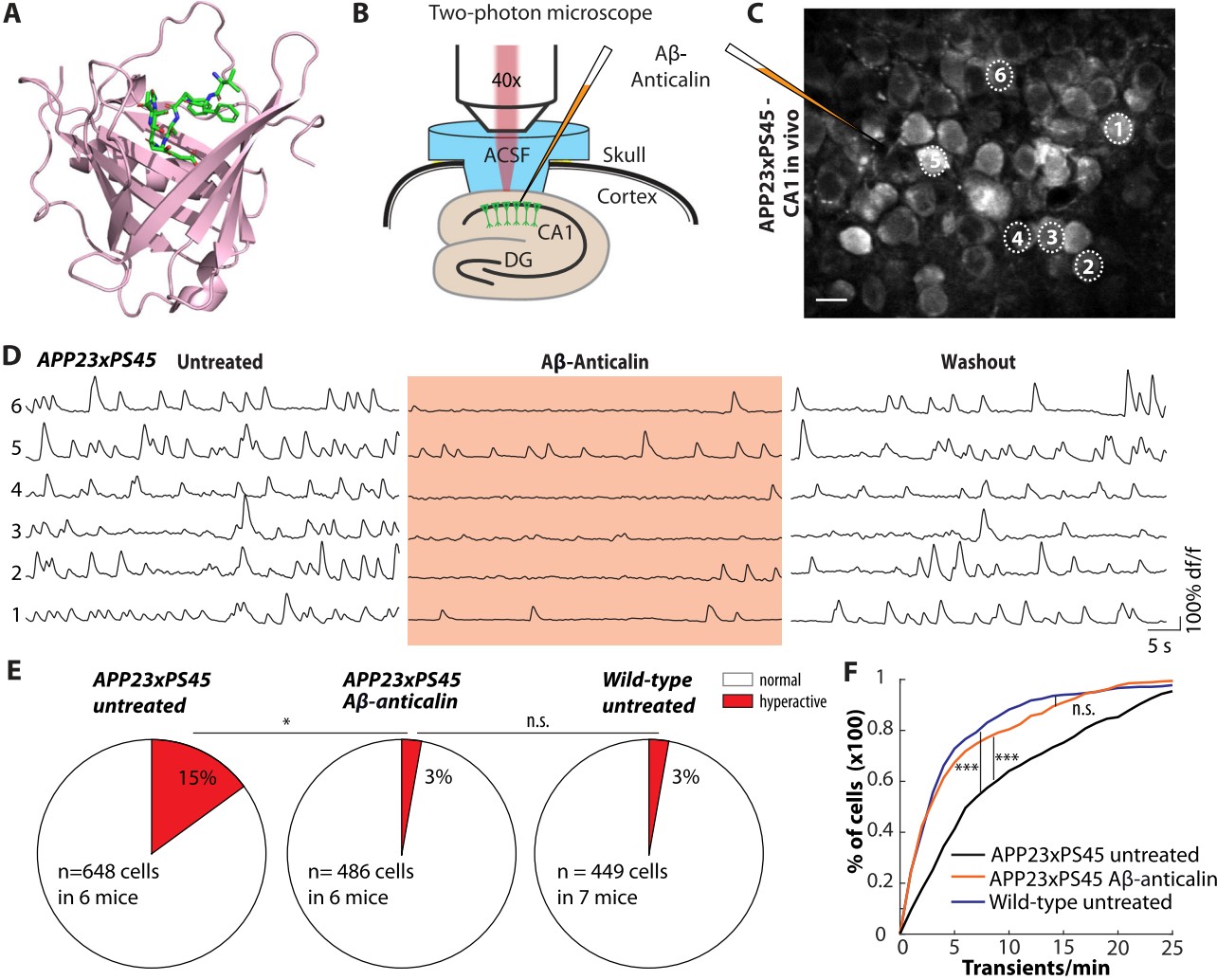

**Fig. 1 | Aβ-anticalin treatment suppresses neuronal hyperactivity in vivo.**
**A** X-ray structure (PDB ID: 4MVL) of the Aβ-anticalin with its β-barrel formed by eight anti-parallel β-strands and four hypervariable loops shown in ribbon presentation. The bound central segment of the Aβ(1-40) peptide is shown as sticks and colored green. **B** Two-photon imaging setup for in vivo imaging of the hippocampal CA1 region. The filled pipette for the application of the Aβ-anticalin is indicated in orange. ACSF, arterial cerebrospinal fluid; CA1, cornu ammonis 1; DG, dentate gyrus. **C** Representative two-photon image of the pyramidal layer of the hippocampal CA1 region in a 2-month-old *APP23xPS45* mouse after staining with the organic Ca²⁺-indicator Cal-520 AM. The injection pipette for the application of Aβ-anticalin is visible as a dark shadow on the left side of the image and

schematically outlined for clarity. Scale bar: 10 μm. **D** Ca²⁺-traces recorded from the six representative neurons labeled in **C** under baseline conditions (*left*), during the application of 10 μM Aβ-anticalin (*middle*) and after a washout period of 5 min (*right*). **E** Percentage of hyperactive cells (more than 20 Ca²⁺-transients per min) in untreated (*left*) and Aβ-anticalin-treated *APP23xPS45* (*middle*) as well as in untreated wild-type animals (right). **F** Cumulative probability of the neuronal activity in untreated (black) and Aβ-anticalin-treated *APP23xPS45* mice (orange) and in untreated wild-type mice (blue). *$p < 0.05$, ***$p < 0.001$, n.s. not significant. Two-sided Wilcoxon signed-rank or rank sum test (E), two-sided Kolmogorow–Smirnow-test (**F**).

the Aβ-anticalin to wild-type mice (Fig. S8, B and D) did not change synaptically evoked glutamate transients. Collectively, these results establish that Aβ scavenging can reverse neuronal hyperactivity and synaptic glutamate accumulation.

**The Aβ-anticalin effect is not dependent on direct interaction with Aβ monomers or dimers**

As an independent test for the anti-Aβ action of the Aβ-anticalin, we explored its effects on neuronal hyperactivity, which was experimentally induced through the direct application of Aβ to the hippocampus of 2–4 month-old wild-type mice[10,18]. For this experiment, we used the synthetic disulfide cross-linked Aβ dimer [AβS26C]₂, which was shown to be a potent neurotoxic Aβ-agent in various assays, including neurite outgrowth[3], the induction of long-term potentiation (LTP)[36,37] and neuronal hyperactivation[10,18]. It is important to note that in all these

assays, applications of [AβS26C]₂ faithfully reproduced the actions of human Aβ extracts, particularly those of natural Aβ dimers.

In line with earlier reports[10,18], we first observed that the local application of [AβS26C]₂ to hippocampal CA1 neurons caused robust neuronal hyperactivation (Fig. 3, A and B). However, in contrast to the hyperactivity-blocking effect in the transgenic mice, the Aβ-anticalin did not have any blocking action on the [AβS26C]₂–induced hyperactivity in wild-type mice (Fig. 3, C and D), and the actions of both applications were undiscernible (Fig. 3, E–G). While this result had to be expected based on the binding specificity of the Aβ-anticalin[24], it is still remarkable as [AβS26C]₂ mimics the action of human Aβ dimers[18]. Thus, the experiment does not fully answer whether binding to monomers or dimers is necessary for the hyperactivity-reducing action of the Aβ-anticalin but may prompt further investigation of the interaction of the anticalin with different forms of Aβ.

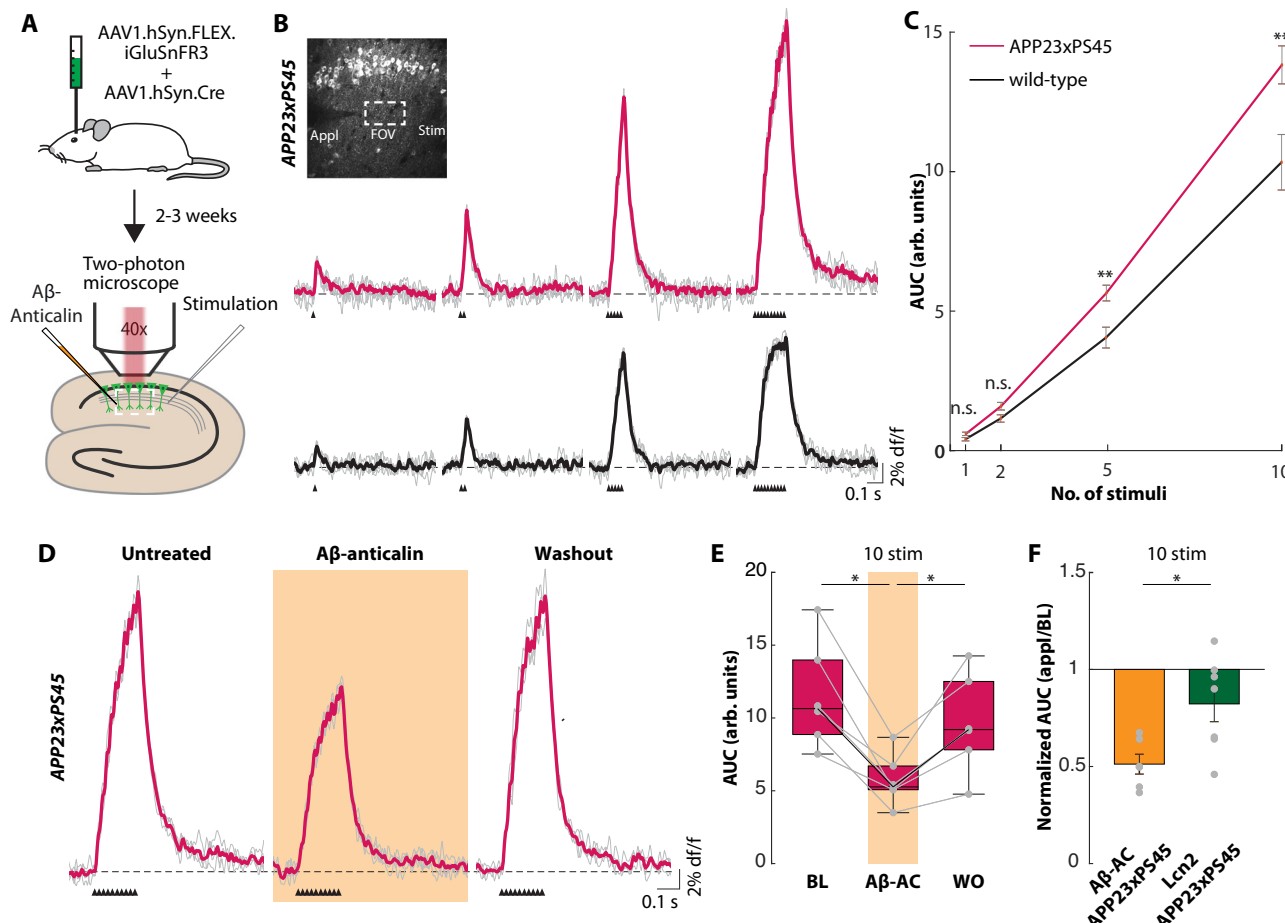

**Fig. 2 | The Aβ-anticalin normalizes glutamate transmission in young *APP23xPS45* mice. A** Schematic depiction of the SF-iGluSnFR-based glutamate imaging experiment. AAV injection in vivo (*top*) followed by in vitro population imaging (*bottom*). **B** Individual synaptically evoked glutamate transients (*gray*) and average (*colored*) from 2-month-old *APP23xPS45* (*top*) or age-matched wild-type (*bottom*) mice for one, two, five and ten stimuli as indicated by black triangles. *Inset*: Two-photon image of iGluSnFR3-expressing CA1 pyramidal neurons and the positions of the pipettes. *Appl*: application pipette, *Stim*: stimulation pipette, FOV: field of view (scale bar: 50 μm). **C** Area under the curve (AUC) of individual gluta-mate transients from wild-type (40 transients in *N* = 8 slices) and *APP23xPS45* (65 transients in *N* = 13 slices) mice for one, two, five and ten stimuli. The error bars depict SEM. **D** Same as (**B**) for ten stimuli in a slice from an *APP23xPS45* mouse

during baseline conditions (*left*), during the application of Aβ-anticalin (10 μM; *middle*) and after washout (*right*). **E** Summary data of the experiments in (**D**) from *N* = 6 slices. Dots represent the mean area under the curve (AUC) of one slice. Box plot is represented as median and quartiles (bounding box). Whisker length is the distance to the furthest observation, but no further than 1.5 times the range from the median to the respective quartile. **F** Comparison of the AUC of glutamate transients during application of Aβ-anticalin (10 μM) or lipocalin 2 (10 μM, *N* = 7 slices) in *APP23xPS45* mice normalized to the respective AUC under baseline conditions (mean +/− SEM). Source data for Fig. 2E, F are provided as a Source Data file. n.s. not significant, *p < 0.05, **p < 0.005. Two-sided Wilcoxon rank sum test (**C**, **F**), two-sided Wilcoxon signed-rank test (**E**).

It is well established that the Aβ-anticalin binds Aβ monomers with high affinity[23,24]. To determine the precise Aβ-binding behavior of the Aβ-anticalin, we performed size exclusion chromatography (SEC) of the Aβ-anticalin alone and after mixture with freshly prepared Aβ monomers. Compared to the isolated Aβ-anticalin (Fig. 4, A and B), we observed a shift of the elution volume after combination with freshly prepared Aβ monomers, corresponding to an increase in size by approximately 4.4 kDa (Fig. 4 C), in line with the known mass of 4430 Da for the Aβ monomer.

In light of the strong affinity for monomers, we asked whether binding Aβ monomers by the Aβ-anticalin directly prevented their neurotoxic actions. In agreement with previous reports that Aβ monomers do not impair LTP[38] or neurite outgrowth[39], as well as the observation that human brain-derived monomers do not cause neu-ronal hyperactivation[18], we observed that the administration of a freshly prepared synthetic Aβ monomer solution in vivo did not cause neuronal hyperactivation in CA1 in 2–4 month-old wild-type mice (Fig. 4, D and E). To exclude a dose-dependent effect, we also per-formed two-photon imaging in acutely prepared hippocampal slices,

in which neuronal in vivo-like baseline activity had been induced by the application of bicuculline (Fig. S9). Under these conditions, the application of 1 μM or 100 μM solutions of the Aβ monomers did not produce neuronal hyperactivation (Fig. 4, F and G).

## The Aβ-anticalin prevents Aβ oligomerization

As Aβ monomers, themselves, did not induce neuronal dysfunction, we hypothesized that scavenging of the (nascent) Aβ monomers by the Aβ-anticalin must reduce the concentration of toxic Aβ oligomers in the brain, either by preventing their formation or by facilitating their disintegration.

To surveil the aggregation behavior of Aβ, we used a Thioflavin T (ThT) fluorescence assay as a probe for the formation of β-sheets[40] that are characteristic for Aβ fibrils. We performed an Aβ 'aging' experi-ment, in which incubated Aβ monomers immediately started forming β-sheet aggregates as demonstrated by a steep initial increase in the ThT-fluorescence, which decayed after approximately two hours, indicating completion of aggregate/fibril formation (Fig. 5A)[41]. The addition of the Aβ-anticalin at the beginning of the incubation period,

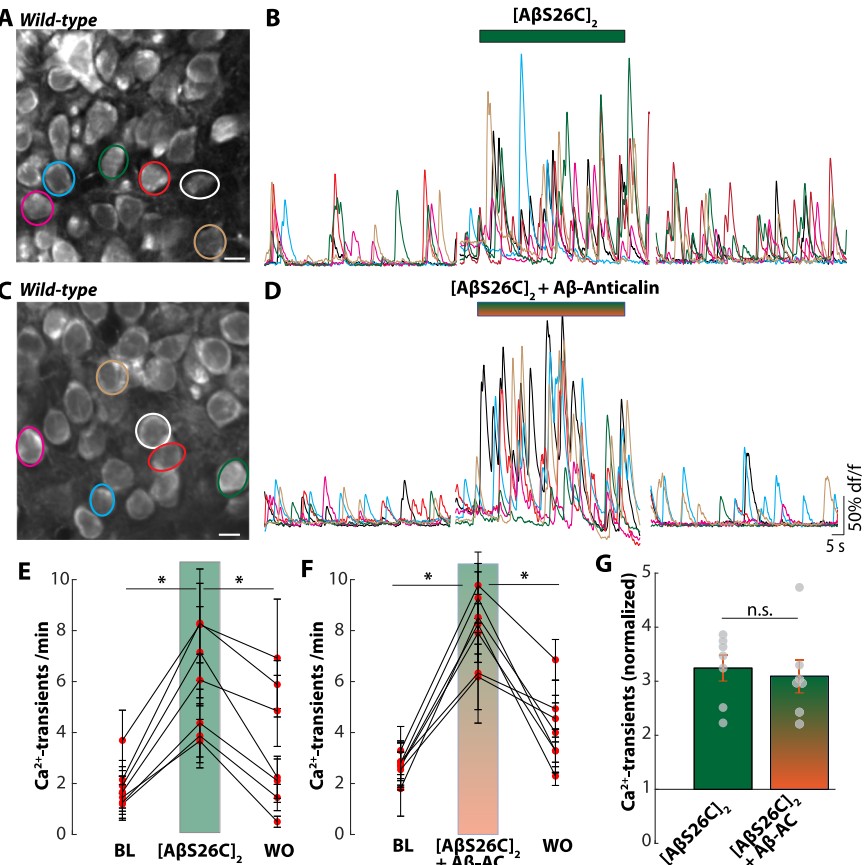

**Fig. 3 | Ineffectivity of the Aβ-anticalin to prevent Aβ-dimer-induced neuronal hyperactivity. A** Representative two-photon image of the pyramidal layer of the hippocampal CA1 region in a 2-month-old wild-type mouse after staining with the organic $Ca^{2+}$-indicator Cal-520 AM. **B** Superimposed representative $Ca^{2+}$-traces of the six neurons cycled in (**A**) under baseline conditions (left), during the application of 500 nM $[AβS26C]_2$ and after 5 min washout. The colors of the $Ca^{2+}$-traces correspond to the circles in (**A**). **C, D** same as (**A** and **B**) for the co-application of $[AβS26C]_2$ (500 nM) and Aβ-anticalin (1 μM). **E** Summary data of the application experiments in (**B**) from $N = 7$ mice. Each dot represents the mean number of $Ca^{2+}$ transients per minute for all observed neurons in one mouse under baseline conditions (BL, *left*), during the application of $[AβS26C]_2$ (*middle*), and after washout (WO, *right*). Data are presented as mean values +/− SEM. (**F**) Same as (**E**) for the experiment in (**D**). **G** Number of $Ca^{2+}$ transients during the application of $[AβS26C]_2$ alone ($N = 7$ mice) or mixed with Aβ-anticalin ($N = 7$), normalized to the respective mean baseline activity (mean +/− SEM). Source data for Fig. 3G are provided as a Source Data file. Scale bars: 5 μm. n.s. not significant. *$p < 0.05$. Two-sided Wilcoxon signed-rank test (**E**, **F**), two-sided Wilcoxon rank sum test (**G**).

however, fully prevented the formation of ThT-positive Aβ fibrils (Fig. 5B), as previously demonstrated by transmission electron microscopy[23]. In contrast, the addition of the Aβ-anticalin after 90 min, i.e. after ThT-positive fibrils had formed, did not reduce fluorescence, indicating that preformed fibrils were not dissolved by the addition of the Aβ-anticalin (Fig. 5C).

Based on the kinetics of Aβ aggregate/fibril formation, we conclude that the freshly prepared synthetic Aβ monomers immediately start forming aggregates and assume that Aβ oligomers should arise as an intermediate step on the way to final fibril formation[42]. To test this, we applied the 'aged' Aβ solution, i.e. Aβ, which had been incubated for 90–120 min (Fig. 5D), to CA1 pyramidal neurons of 2–4 month-old wild-type mice in vivo, which robustly induced neuronal hyperactivation (Fig. 5, E and J), in contrast to the observed ineffectiveness of the freshly prepared monomer solution (Fig. 4). Likewise, 'aged' Aβ application also caused neuronal hyperactivation in hippocampal slices, both at concentrations of 1 and 100 μM monomer equivalent (Fig. S10, A to D).

Given that the ThT-fluorescence assay indicates that the Aβ-anticalin prevents the formation of Aβ fibrils (Fig. 5B), we asked whether the Aβ-anticalin could also prevent the formation of toxic soluble Aβ dimers and oligomers. Thus, we next administered the solution of Aβ monomers, which had been incubated in vitro with a stoichiometric concentration (1:1) of the Aβ-anticalin for 90–120 min

(Fig. 5F). The application of this solution in vivo did not induce neuronal hyperactivation (Fig. 5, G and K), demonstrating, first, that toxic Aβ dimers or oligomers had not formed and, second, that anticalin-bound Aβ monomers do not induce neuronal hyperactivation (Fig. 5L). Likewise, the application of this solution did not change the neuronal activity levels in bicuculline-treated hippocampal slices (Fig. S8, E and F).

Next, we asked whether the neutralizing effect of the Aβ-anticalin might at least partially be mediated by the disintegration of neurotoxic Aβ aggregates. When we applied the mixture of 'aged' Aβ and Aβ-anticalin (Fig. 5C) in the hippocampal CA1 region in vivo, it induced neuronal hyperactivation (Fig.5, I and M), which was undiscernible from that caused by the Aβ oligomers alone (Fig. 5N). In addition to the increased number of $Ca^{2+}$-transients, we analyzed the AUC for the respective $Ca^{2+}$-signals (Fig. S11). We found that the application of aged Aβ alone or together with the Aβ-anticalin reliably increased the AUC by a similar percentage. In contrast, the application of freshly prepared Aβ monomers alone or incubated with the Aβ-anticalin had no effect. In line with this, in the SEC analysis of the mixture between aged Aβ and the Aβ-anticalin, we found a slight shift to a lower elution volume (in line with larger size by ~2.7 kDa) as well as a broadening of the peak compared to the anticalin protein alone (Fig. S12). This indicates the binding of residual Aβ monomers but not of larger oligomers or aggregates to the Aβ-anticalin.

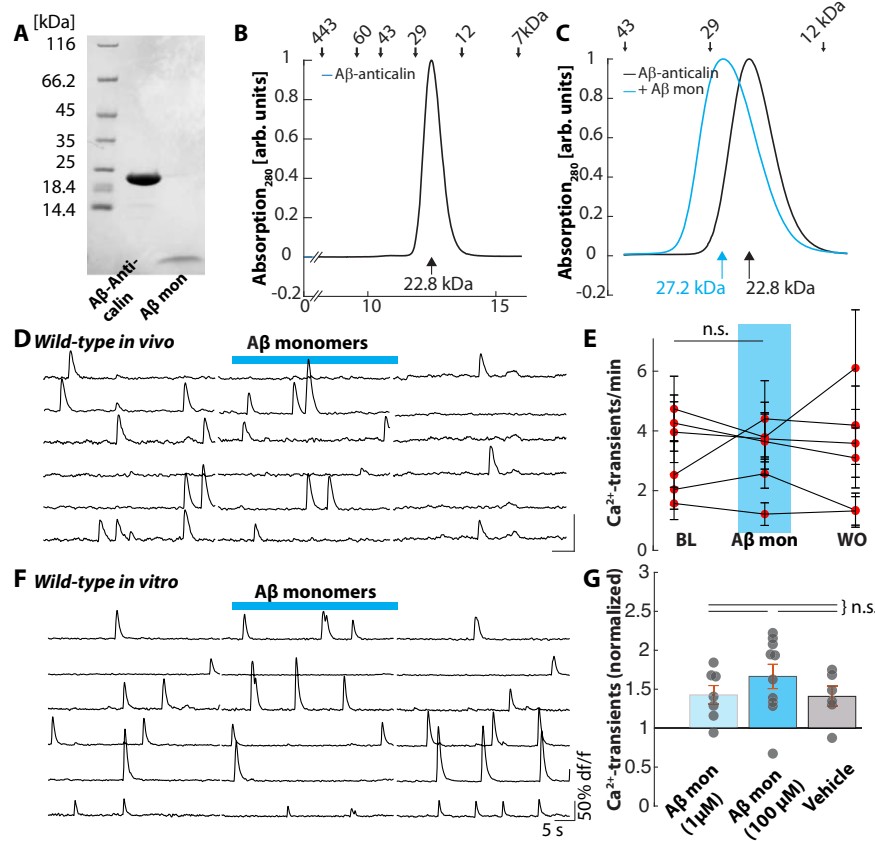

**Fig. 4 | Aβ-anticalin binds inactive Aβ monomers. A** Coomassie-stained SDS/PAGE depicting the size of Aβ-anticalin (21.3 kDa) and Aβ monomer (4.4 kDa). The experiment was repeated twice. **B** Size exclusion chromatography (SEC) of the Aβ-anticalin shows a peak at an elution volume of 12.48 ml, corresponding to an apparent molar weight of 22.7 kDa. **C** SEC of freshly prepared Aβ(1-40) monomers (calculated molar weight: 4430 Da) in combination with Aβ-anticalin demonstrate a shift of the peak from 12.48 to 12.07 ml, suggesting binding of the 4.4 kDa-peptide by the anticalin. **D** Representative Ca²⁺-traces recorded from six hippocampal CA1 neurons of a wild-type mouse under baseline conditions, during the application of Aβ monomers (10 μM, applied 5 min after preparation of the solution) and after a washout period of 5 min. **E** Summary data of the experiments in (D) from N = 6 mice. Each dot represents the mean number of Ca²⁺- transients per minute, for all observed neurons in one mouse under baseline conditions (BL), during the application of Aβ monomers (Aβ mon) and under washout conditions (WO). **F** Ca²⁺-transients from six representative CA1 pyramidal neurons in a bicuculline-treated hippocampal slice from a wild-type mouse under baseline conditions (*left*), during the application of Aβ(1-40) monomers (1 μM, applied 5 min after preparation of the aqueous solution) through a patch pipette (*middle*) and after washout for 5 min (*right*). **G** Number of Ca²⁺-transients during the application of 1 μM Aβ monomers (N = 6 slices), 100 μM Aβ monomers (N = 10) or ACSF as vehicle solution (N = 6), normalized to the respective mean baseline activity. n.s. not significant. Source data for Fig. 4G are provided as a Source Data file. Error bars depict SEM. Two-sided Wilcoxon signed-rank test (E) or Kruskal–Wallis test with Dunn-Sidac post-hoc comparison (G).

In light of the apparent lack of binding to aggregated Aβ-species, it is important to ask whether Aβ-anticalin treatment can also be effective in older *APP23xPS45* mice with a large number of plaques (Fig. S13A). While old *APP23xPS45* mice also had a high percentage of hyperactive neurons (Fig. S13B), which was similar to that of young animals, the application of the Aβ-anticalin only marginally reduced neuronal activity levels in 7–8 month-old plaque-bearing AD animals (Fig. S13, C and D). Likewise, the application of the γ-secretase inhibitor had a markedly smaller effect on preventing neuronal hyperactivity in the aged AD mice compared to young mice (Fig. S10 E, F).

Finally, we asked whether the Aβ-anticalin dependent prevention of Aβ aggregation also prevented perisynaptic glutamate accumulation. To this end, we expressed the fluorescent glutamate indicator SF-iGluSnFR A184S in the CA1 region of wild-type mice (Fig. 6A, B) and synaptically induced extracellular glutamate transients (Fig. 6C, *left*) by electric stimulation of the Schaffer collaterals. The application of 'aged' Aβ-solution (Fig. 5A) caused a robust increase in the synaptically evoked glutamate transients (Fig. 6C, *middle*), which was reversible after a washout of 5–10 min (Fig. 6C, *right*). These results were similar to previous results with [AβS26C]₂[18]. In contrast, the application of a similarly 'aged' solution of an Aβ(40-1) peptide with reverse sequence,

which does not form aggregates[43], did not affect synaptically evoked glutamate transients (Fig. 6D). In a parallel approach, injection of Aβ(1-40) solution incubated with the Aβ-anticalin (Fig. 5B), putatively containing bound Aβ monomers, did not enhance synaptically evoked glutamate transients (Fig. 6, E and F). Together, these data demonstrate that preventing the formation of toxic Aβ oligomers or aggregates also prevents the synaptic dysfunction underlying Aβ-induced neuronal hyperactivity.

## Discussion

In this study, we demonstrate that scavenging Aβ monomers by an Aβ-anticalin can restore normal neuronal activity levels in mouse models of AD and provide mechanistic evidence for its rescue action. Under disease conditions, naturally secreted Aβ monomers, which are pathologically inactive, rapidly form toxic dimers and oligomers that cause neuronal hyperactivity in vivo. However, adding the Aβ-anticalin at the beginning of the aggregation cascade scavenges Aβ monomers, thereby preventing the formation of toxic aggregates. In consequence, neuronal activity levels remain unaltered (Fig. 6G).

Hippocampal hyperactivity is an early neuronal dysfunction of AD in mice and humans, precedes plaque deposition as well as overt brain

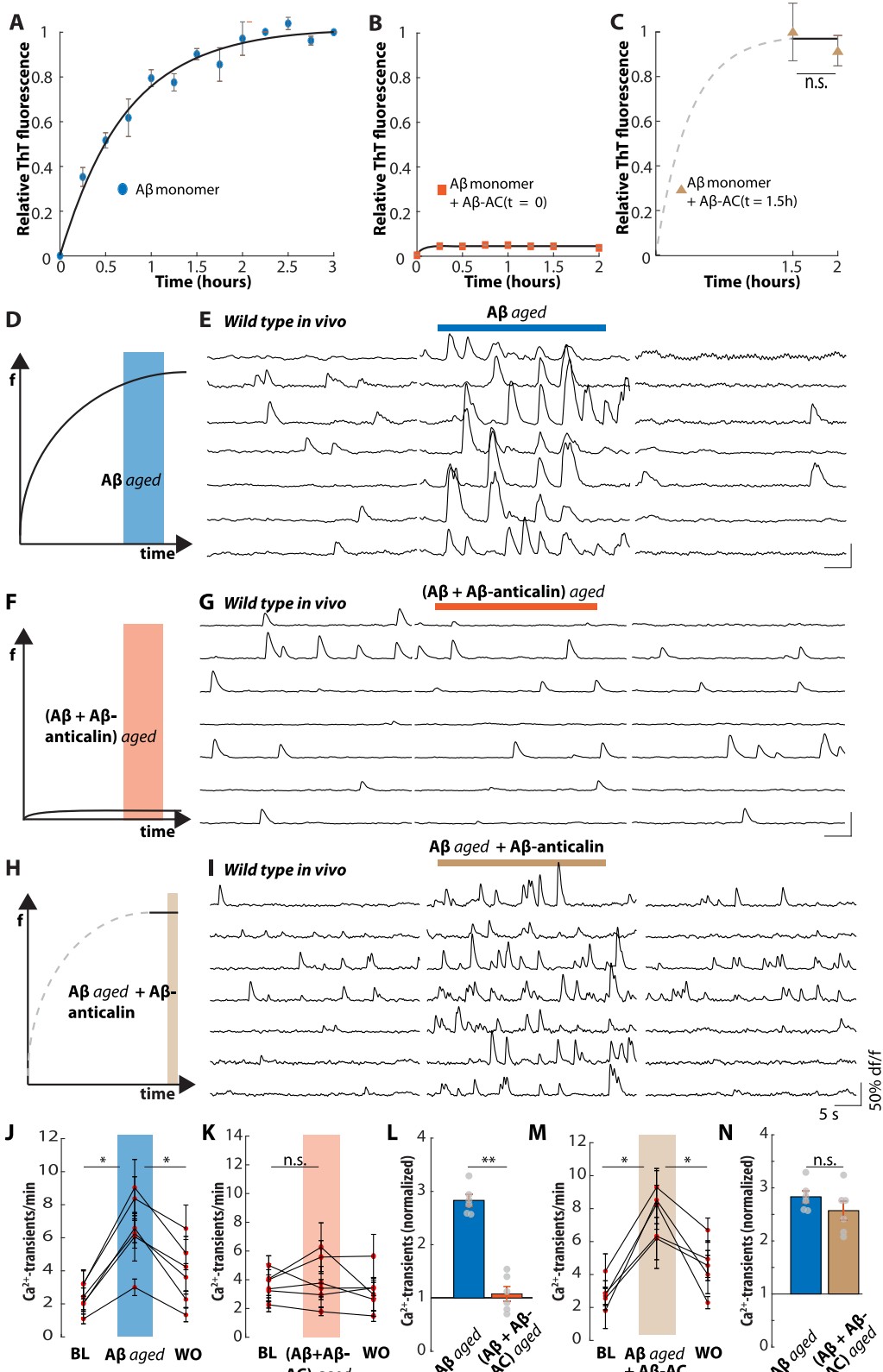

atrophy[17,44], and is associated with the break-down of circuits and impaired cognition[45,46]. Moreover, studies in mice and humans demonstrated that the effective suppression of neuronal hyperactivity can improve or even restore cognitive function[20,47]. Neuronal hyper-activity can be prevented or reduced in mouse models by blocking Aβ secretion[10,20], but previous attempts at scavenging Aβ using parenteral application of mAbs have even exacerbated neuronal dysfunction[6].

Consequently, our findings that the direct intracerebral application of the Aβ-anticalin and, to a certain degree, also Solanezumab can restore normal neuronal activity, demonstrate the feasibility of such approaches.

Moreover, we provide direct evidence using fluorescent gluta-mate indicators that scavenging of Aβ by the Aβ-anticalin prevents extracellular glutamate accumulation, a prominent feature of AD,

**Fig. 5 | Prevention of the formation of toxic aggregates of the Aβ-anticalin. A** In vitro aggregation assay of synthetic Aβ(1-40) monomers. Average (*n* = 3 experiments) and SEM are shown. The values were normalized to the maximum fluorescence. **B** same as (**A**) for samples taken from the Aβ aggregation assay in the presence of the Aβ-anticalin (Aβ-AC). **C** Same as (**B**) after aggregation for 90 min and following the addition of the Aβ-anticalin **D** Scheme depicting the aggregation curve of Aβ(1-40) monomers in the ThT assay. f, fluorescence (**E**) Representative Ca²⁺-traces recorded from seven hippocampal CA1 neurons of a wild-type mouse during baseline, during the application of putative Aβ oligomers (10 μM monomer equivalent) and after washout. **F** Scheme depicting the ThT-aggregation assay in the presence of Aβ and the Aβ-anticalin (10 μM, respectively). **G** Same as (**E**) for the application of putative protein-bound Aβ monomers. **H** Scheme depicting the aggregation of Aβ monomers alone for 90 min and after the addition of Aβ-

anticalin. **I** Same as (**E**) for the application of 'aged' Aβ (10 μM monomer equivalent), incubated with 10 μM Aβ-anticalin for a further 30 min. **J** Summary data of the experiments in (**E**) from *N* = 6 mice under baseline conditions (BL), during the application of 'aged' Aβ and after washout (WO). **K** same as (**J**) for the experiment in (**G**) and the application of Aβ/Aβ-anticalin. **L** Number of Ca²⁺-transients during the application of 'aged' Aβ (*N* = 6) or Aβ/Aβ-anticalin (*N* = 6), normalized to the respective mean baseline. **M** same as (**J**) for the experiment in (**I**) and the application of 'aged' Aβ, incubated with Aβ-anticalin. **N** Number of Ca²⁺-transients during the application of 'aged' Aβ (*N* = 6) or 'aged' Aβ, incubated with Aβ-anticalin (*N* = 6), normalized to the respective mean baseline activity. Data in 5C and 5J–N are presented as mean +/− SEM. Source data for Fig. 5L and N are provided as a Source Data file. Data in n.s. not significant, *\*p* < 0.05, *\*\*p* < 0.005, Two-sided Wilcoxon signed-rank test (**J, K, M**), Two-Wilcoxon rank sum test (**C, L, N**).

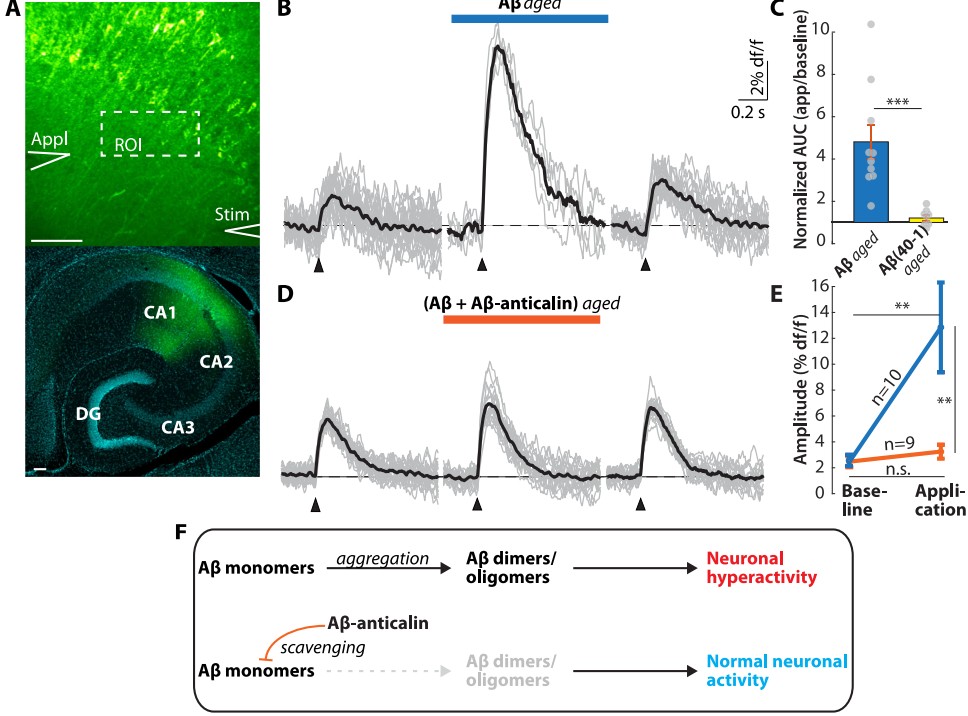

**Fig. 6 | The Aβ-anticalin prevents Aβ-dependent synaptic dysfunctions. A** *Top:* representative two-photon image of the SF-iGluSnFRA184S-expressing CA1 region of a wild-type mouse. The locaton of the application (*Appl*) and the stimulation (*Stim*) pipette, as well as the region of interest (ROI) from which glutamate transients were recorded, are outlined in white. *Bottom:* post-hoc confocal image of the SF-iGluSnFr-expressing neurons in CA1 (*green*) and DAPI counterstaining (*cyan*). DG: dentate gyrus, CA: cornu ammonis. Scale bars: 100 μm. **B** Individual synaptically evoked glutamate transients (*gray*) and average (*black*) from one slice during baseline conditions (*left*), during the application of 'aged' Aβ (50 μM monomer equivalent, incubated for 20 min in ACSF; *middle*) and after washout (*right*). **C** Area under the curve (AUC) of the evoked glutamate transients during the application of 'aged' Aβ(1-40) (*N* = 10 slices) or 'aged' reverse Aβ(40-1) (*N* = 9), normalized to the respective AUC registered under baseline conditions. Data are presented as mean

values +/- SEM. **D** Same as (**C**) for the application of Aβ + Aβ-anticalin (50 μM Aβ monomer equivalent, incubated in ACSF containing 50 μM of the Aβ-anticalin for 20 min). **E** Comparison of the amplitudes of glutamate transients before (*left*) and during (*right*) the application of 50 μM equivalent Aβ monomer, incubated for 20 min without (*blue*) or with (*orange*) Aβ-anticalin (mean +/- SEM). **F** Scheme depicting the mechanism of action of the Aβ-anticalin. Under disease conditions, nascent Aβ monomers rapidly aggregate into toxic dimers and oligomers, which cause neuronal hyperactivity (*top*). Scavenging these monomers by the Aβ-anticalin disrupts the formation of toxic dimers/oligomers, thus preventing neuronal hyperactivation (*bottom*). Source data for Fig. 6C are provided as a Source Data file. n.s. not significant, *\*\*p* < 0.005, *\*\*\*p* < 0.001. Two-sided Wilcoxon rank sum test (**C, E** across groups); Wilcoxon signed-rank test (**E**, baseline vs. application).

which is associated with impaired synaptic integrity and plasticity as well as a mechanism of neuronal hyperactivity[18,31,33,48]. Our glutamate imaging data in hippocampal slices demonstrates that the extent of the perisynaptic glutamate accumulation is correlated with the amount of synaptic stimulation. This is in line with our previous findings that Aβ-dependent neuronal hyperactivation requires ongoing synaptic input[18]. Our data provide a mechanistic explanation for the reduced neuronal hyperactivity observed in our in vivo experiments.

In line with previous reports[18,38,39], we found that Aβ monomers are pathologically inert. Thus, solutions containing the freshly

prepared Aβ monomers only or Aβ monomers stably bound by the Aβ-anticalin failed to induce neuronal hyperactivity both in vivo and in vitro and, furthermore, did not lead to extracellular glutamate accumulation. On the contrary, a solution of aggregated, i.e. 'aged', Aβ including dimers and oligomers, potently perturbed neuronal function. These observations reinforce previous findings that such oligomers are the most toxic species of Aβ[18,36,49,50].

In light of this, the rapidity of the hyperactivity-reducing effect of the Aβ-anticalin application in vivo is remarkable because the brains of *APP23xPS45* mice already contain elevated levels of oligomeric or

aggregated Aβ[20]. Thus, our data suggest that neuronal dysfunction in early AD largely depends on the continuous de novo release of (nascent) Aβ monomers, which immediately start forming toxic oligomers or aggregates, rather than a stable pool of such Aβ species. In consequence, it is likely that existing oligomers are quickly rendered innocuous in vivo, either by clearance from the brain or by the formation of larger, less toxic aggregates[50]. Additional evidence for this hypothesis comes from a parallel experiment with application of a γ-secretase inhibitor, which had a very similar effect as the Aβ-anticalin in our in vivo hyperactivity assay. In line with this, previous studies have shown that systemic γ-secretase inhibitor treatment prevents the release of Aβ peptides into the extracellular space[51], leads to a rapid decrease of total[30,52] and oligomeric[10] Aβ levels in mouse models of β-amyloidosis and, concomitantly, normalized neuronal activity[10]. While previous findings did not exclude that oligomers can form intracellularly before they are released[53–55], based on the efficacy of the Aβ-anticalin in *APP23xPS45* mice, it is unlikely that such released oligomers reach sufficiently high concentrations to cause neuronal dysfunction in vivo.

Taken together, our data indicate that, in young AD mice, monomer scavenging by the Aβ-anticalin or Solanezumab can prevent Aβ-induced neurotoxic effects by scavenging the monomeric peptide and interfering with its aggregation without the need or ability to dissolve preexisting oligomers.

In older, plaque-bearing *APP23xPS45* mice, on the other hand, scavenging monomers alone was not sufficient to reduce neuronal hyperactivity, possibly because oligomers are not only formed from nascent Aβ monomers in these aged animals but can also be released from larger aggregates[50,56]. However, our data are compatible with the possibility that Aβ-anticalin treatment may be a valuable addition to therapies aiming at the removal of large aggregates.

The current study demonstrates that Aβ-anticalin dependent Aβ monomer scavenging can repair neuronal and synaptic dysfunctions in mouse models of AD in vivo. However, the main limitation of our approach is that, by the direct and acute injection of the Aβ-anticalin in a defined small brain area, we were only able to investigate local short-term effects of the application. In consequence, more work is needed to clarify whether Aβ-anticalin application may also constitute a promising therapeutic strategy and can mitigate plaque pathology and cognitive decline longitudinally. Their small size and efficacy at relatively low concentrations, and higher efficacy over the antibody Solanezumab in our assay, together with a lack of immunological effector functions as well as interactions with plasma proteins[24], make the Aβ-anticalin a potential candidate drug for further investigation as a possible therapeutic entity for the preventive treatment of early AD.

## Methods

All experimental procedures were performed in compliance with all relevant ethical regulations and were explicitly approved by the government of upper Bavaria (animal protocol numbers: 55.2-1-54-25323-11, 55.2.1.54-2532-66-13, ROB-55.2-2532.Vet_02-16-155, ROB-55.2-2532.Vet_02-21-121).

### Animal models

In vivo experiments were performed with 2–4 month-old (young group) and 7–9 month-old (old group) C57 Bl/6 N wild-type mice of both sexes or with age-matched female *APP23xPS45* mice expressing the APP Swedish mutation (670/671) and the G384A mutation in the presenilin 1 (PS1) gene under the Thy-1 promotor[14]. Additionally, we used 4–8 month-old *APP23* mice of both sexes[57]. All mice were housed in standard mouse cages under a 12-h dark/12-h light cycle and constant temperature ( ~ 25 °C) and humidity ( ~ 55%). Food and water were provided ad libitum.

### Surgery

In vivo two-photon imaging was performed in line with previous reports[10,18]. Mice were initially anesthetized with isoflurane (2% vol/vol in pure $O_2$). The scalp was partially removed and a custom-made plastic recording chamber with a central opening was attached to the skull using dental cement. The skull was thinned with a dental drill (Meisinger, Neuss, Germany) in a circle with a diameter of approximately 2 mm with the center on top of the hippocampal CA1 region (AP −2.75, ML 3.5) and the recording chamber was filled with artificial cerebrospinal fluid (ACSF; 125 mM NaCl, 4.5 mM KCl, 26 mM NaHCO₃, 1.25 mM NaH₂PO₄, 2 mM CaCl₂, 1 mM MgCl₂, 20 mM glucose, pH 7.4 when bubbled with carbogen gas), which had been warmed to 37 °C. The bone was carefully removed with a thin cannula to open a cranial window directly above the imaged region. The dura and cortical tissue covering the hippocampus were removed by suction. After this, multi-cell bolus loading was performed with the organic $Ca^{2+}$-indicator Cal-520 AM[58], injected through a glass patch pipette 200 μm underneath the hippocampal surface. After the surgery, the concentration of Isoflurane was reduced to 0.8–1.0% vol/vol for imaging.

### Anticalin preparation and antibodies

The Aβ-specific Anticalin was prepared via soluble cytoplasmic expression in 2 L shake flask cultures of *E. coli* Origami B[59]. Wild-type Lcn2 was periplasmatically expressed in *E. coli* W3310[60,61]. Both proteins were purified by His₆-tag affinity chromatography[62] and SEC on a Superdex 75 HR 26/60 column (GE Healthcare, Munich, Germany) using HEPES-Ringer (135 mM NaCl, 5 mM KCl, 2 mM CaCl₂, 1 mM MgCl₂, 10 mM HEPES, 20 mM Glucose, pH 4.7) or phosphate-buffered saline (PBS; 4 mM KH₂PO₄, 16 mM Na₂HPO₄, 115 mM NaCl, pH 7.4). Protein purity was checked by SDS/PAGE[63] and ESI mass spectrometry on an maXis instrument (Bruker Daltronics, Bremen, Germany) in the positive ion mode. Endotoxins were removed using NoEndo™ HC (High Capacity) Spin Columns (Protein Ark, Rotherham, UK) and residual pyrogen levels were analyzed using the Endosafe-PTS™ (Charles River Laboratories, Wilmington, USA) resulting in < 5 EU/mL. Protein concentration was determined via absorption at 280 nm using molar absorption coefficients calculated with the ExPASy ProtParam tool[64].

The IgG Solanezumab biosimilar antibody was obtained from Antibodies-online (Cat.# ABIN7487922, CAS 955085-14-0), the Goat anti-Rabbit IgG unconjugated control antibody was purchased from Invitrogen (Cat.# A27033). Proteins were stored at −20 °C and diluted to the end concentration of 1 μM in ACSF or HEPES-Ringer immediately before the experiment.

### Aβ peptides

Aβ(1-40) and Aβ(40-1), the latter with a reverse amino acid sequence, were obtained from Bachem Pharmaceuticals (Bubendorf, Germany). The Aβ(1-40/S26C) dimer [AβS26C]₂ was from jpt (Berlin, Germany). All peptides were dissolved in DMSO and stored frozen. Immediately before the experiments, the peptides were thawed and diluted in ACSF or HEPES-Ringer or in Ringer solution containing an equimolar amount of the Aβ-anticalin. The solution was pressure applied through a glass patch pipette either immediately or incubated at room temperature (25 °C) for the indicated time and centrifuged at ~4400 g for 5 min to remove insoluble aggregates before application.

### Hippocampal slice preparation

Mice were anesthetized and decapitated. The brain was surgically removed and submerged in ice-cold slicing solution (24.7 mM glucose, 2.48 mM KCl, 65.47 mM NaCl, 25.98 mM NaHCO₃, 105 mM sucrose, 0.5 mM CaCl₂, 7 mM MgCl₂, 1.25 mM NaH₂PO₄, 1.7 mM ascorbic acid) with an osmolarity of 290–300 mOsm and a pH of 7.4, which was stabilized by bubbling with carbogen gas. Horizontal slices (300 μm)

were cut using a vibratome. These were allowed to recover at room temperature for at least one hour in a recovery solution containing 2 mM CaCl$_2$, 12.5 mM glucose, 2.5 mM KCl, 2 mM MgCl$_2$, 119 mM NaCl, 26 mM NaHCO$_3$, 1.25 mM NaH$_2$PO$_4$, 2 mM thiourea (Sigma, St. Louis, USA), 5 mM Na-ascorbate (Sigma), 3 mM Na-pyruvate (Sigma), and 1 mM glutathione monoethyl ester. The pH was adjusted to 7.4 with HCl, and the osmolarity was 290 mOsm. Before the imaging experiment, the Schaffer collaterals were cut and the slices were transferred into the recording setup and superfused with warmed (37 °C) ACSF. To induce in vivo-like ongoing baseline activity in the hippocampal slices, 100 μM bicuculline was added to the ACSF and the potassium concentration was slowly increased to 5.5–6.5 mM (for a detailed description, see[18]). For calcium imaging experiments, bolus loading of Cal-520 AM was performed as described above.

## Two-photon Ca$^{2+}$-imaging and application of Aβ and/or anticalins

In vivo and in vitro two-photon Ca$^{2+}$ imaging was performed in the same custom-made multi-photon recording setup based on an upright microscope (Olympus, Shinjuku, Japan)[18]. Excitation light was provided by a tunable Ti:sapphire laser at a wavelength of 920 nm (Coherent, Santa Clara, USA). Fluorescence images were collected using a resonant galvo mirror scanner operating at 8 or 12 kHz (GSI group, Bedford, USA) as well as a 40 × 0.8 NA objective (Nikon, Tokyo, Japan). Full frames were acquired at 40 Hz.

After surgery or slice preparation and Cal-520 AM bolus loading (see above), spontaneous Ca$^{2+}$-transients were recorded from the pyramidal layer of hippocampal CA1. For the injection of Aβ and/or anticalins a glass patch pipette (tip resistance 1–2 MΩ) was filled with approximately 5 μl of the peptide/protein application solution (see above) and the tip was positioned in the pyramidal layer of the hippocampal CA1 region under visual control (about 150 μm under the hippocampal surface in vivo or 50 μm below the slice surface in vitro). The pressure was carefully applied using a picospritzer II (Parker, Cleveland, USA) until a slight tissue displacement in front of the tip indicated fluid ejection (typically around 20 mBar). In some experiments, this was further validated by adding 5 μM Alexa 594 dye (Thermo Fischer, Waltham, USA) to the pipette solution. The pressure was stopped after 30–60 s. The same field of view was monitored during baseline, pressure application, and washout conditions.

## Virus injection and glutamate imaging

AAV2/1.hSynapsin.iGluSnFR A184S[65] was a gift from Loren Looger (HHMI/Janelia), AAV1.hSyn.FLEX.iGluSnFR3.v857.PDGFR.codonopt and AAV1.hSyn.Cre.WPRE.hGH were purchased from Addgene (Cat#175180-AAV1 and Cat#105553-AAV1). Virus injection and two-photon population glutamate imaging was performed in a similar fashion as previous reports[18,35]. Thus, 500 nl of AAV2/1.hSynapsin.iGluSnFR A184S (2.4×10$^{12}$ gc/ml) or AAV1.hSyn.FLEX.iGluSnFR3.v857.PDGFR.codonopt (3.5 ×10$^{12}$ gc/ml) and AAV1.hSyn.Cre.WPRE.hGH (7.3 ×10$^{11}$ gc/ml) were injected into the hippocampal CA1 region (AP −2.75, ML 3.5 and DV 2–3 mm) of isoflurane-anesthetized 4–6 week-old mice by slow (10–20 nl/min) pressure injection from a glass pipette. After the retraction of the injection pipette, mice were transferred back to their home cage, where 2–3 weeks were allowed for viral expression.

After that, hippocampal slices were prepared as described further above or, for the experiments in Fig. 6, according to the Optimized N-Methyl-D-glucamine Protective Recovery Method[66] with sodium spiking at 34 °C. Glutamate two-photon imaging was performed in the stratum radiatum of the hippocampal CA1 area under illumination at 920 nm with a framerate of 120 Hz for iGluSnFR.A184S and 200 Hz for iGluSnFR3. For experiments with the iGluSnFR3 construct a deformable mirror (DM97-15, Alpao, Montbonnot, France) was introduced in the light path and different Zernicke modes were applied to optimize

image resolution and contrast for each experiment. Synaptic glutamate release was achieved by electrical stimulation of the Schaffer collaterals in CA1. In Fig. 6, single action potentials were evoked with a glass pipette (30–40 V, 100 μs). In Fig. 2, trains of action potentials were evoked by a concentric bipolar electrode (FHC, Bowdoin, USA, Cat.# 30201) at 12–14 V, 100 μs pulse duration and 20 ms interpulse interval. Aβ peptide and/or anticalin were pressure applied in the field of view through a second glass pipette.

## Image analysis

Offline image analysis was performed as described previously[67]. Fluorescence traces were extracted from the imaging data using custom-written software based on LabVIEW. In Ca$^{2+}$-imaging experiments, neurons were visually identified and regions of interest (ROIs) were drawn around their somata. Astrocytes were excluded from the analysis due to their morphology and their high fluorescence levels[68]. In glutamate imaging, a region of interest (ROI) was drawn over the full imaging frame. The fluorescence of each ROI over time was extracted, low-pass filtered to 10 Hz (Ca$^{2+}$) or 40 Hz (glutamate) and normalized to the baseline as $df/f = (f(t) - f0)/f0$, where $f0$ was set at the 10$^{th}$ percentile of the entire trace in Ca$^{2+}$-imaging or at the mean of the pre-stimulus interval in glutamate imaging experiments. Fluorescence changes with peak amplitudes three times larger than the standard deviation of the baseline were accepted as neuronal calcium or extracellular glutamate transients. To calculate the area under the curve (AUC) in Fig. S11, a time-dependent baseline was determined for each cell[67] to prevent confounding effects by slow drifts, especially during application periods. The AUC was calculated individually for each neuron using the trapezoidal rule during 40 s of baseline and the first 40 s of the respective application. For glutamate transients, the AUC was calculated using the trapezoidal rule within 0.5 s after stimulation.

## Measurement of binding activity using surface plasmon resonance (SPR)

Kinetic affinity data of the Aβ-anticalin (H1GA) and the recombinant wild-type lipocalin 2 (Lcn2) were measured at 25 °C on a BIAcore 2000 system (BIAcore, Uppsala, Sweden) with immobilized Aβ(1-40) (Bachem) according to a published procedure[23] using HBS-T (20 mM HEPES/NaOH pH 7.5, 150 mM NaCl and 0.005% v/v Tween20) as running buffer. ~350 RU of Aβ(1-40) was covalently immobilized on a CM5 chip (GE Healthcare) using amine coupling chemistry in 10 mM sodium acetate buffer pH 5. Dilution series of the purified lipocalin proteins were applied at a flow rate of 30 μl/min. The data were double-referenced by subtraction of the corresponding signals measured for the control channel and of the average of three buffer injections[69]. Kinetic parameters were determined by global fitting of single-cycle kinetics with BIAevaluation software v 4.1 using the Langmuir 1:1 binding model. The equilibrium dissociation constants were calculated as $K_D = k_{off}/k_{on}$ and the statistical error was estimated as previously described[70].

## Analytical size exclusion chromatography

2.5 μl of a 5 mM Aβ(1-40) monomer solution in DMSO was mixed with 247.5 μL of 50 μM Aβ-anticalin solution in HEPES-Ringer, resulting in a 1:1 molar ratio of Aβ-anticalin to Aβ. In a control experiment, only the buffer without the Aβ-anticalin was used. 100 μL of the sample was cleared from aggregates by centrifugation in a bench top centrifuge at -20,000 g (5 min, room temperature) and directly applied to SEC. The remaining sample was incubated for 90 min at room temperature, again followed by SEC after centrifugation. The chromatography was performed on a 24 ml Superdex 75 10/300 GL column (GE Healthcare) using HEPES-Ringer buffer pH 7.4 at a flow rate of 0.5 ml/min. The column was calibrated with the following protein standards (Sigma–Aldrich): alcohol dehydrogenase (ADH, 150 kDa) bovine serum

albumin (BSA, 66 kDa), ovalbumin (43 kDa), carbonic anhydrase (CA, 29 kDa), cytochrome c (Cyt c, 12.4 kDa) and aprotinin (Ap, 6.5 kDa). Blue dextran was applied to determine the void volume of the column. Based on the peak elution volumes, the partition coefficients ($K_{av}$) were calculated and used to interpolate the apparent molecular masses of the Aβ/anticalin mixtures.

## ThioflavinT fluorescence assay

The ThT Assay was carried out according to previously published procedure[23]. Aβ(1-40) was lyophilized and dissolved in 1,1,1,3,3,3-hexa-fluoro-2-propanol (HFIP; Sigma–Aldrich) at a concentration of 5 mg/ml. Following the evaporation of HFIP in the fume hood for 12 h, the dried Aβ(1-40) was dissolved in 250 μl ice-cold $H_2O$. After sonication for 15 min at 4 °C (Sonorex, Bandelin, Berlin, Germany), the peptide solution was sterile-filtrated with a Costar Spin-X centrifuge tube filter, 0.45 μm pore size cellulose acetate membrane (Corning Life Sciences, Kaiserslautern, Germany). The freshly prepared solution of monomeric Aβ(1-40) was immediately used for the aggregation assay by mixing the 2 mg/ml (462 μM) Aβ(1-40) monomer solution with either 250 μl Aβ-anticalin solution in PBS at a 1:1 molar ratio or PBS alone. Aggregation reactions were performed in triplicates at 25 °C or 37 °C in 2 ml DNA LoBind Tubes (Eppendorf, Hamburg, Germany) with stirring at 500 rpm using a 5 mm magnetic bar. For fluorescence measurements, 20 μl samples were removed at distinct time points and mixed with 180 μl of a 55.6 μM solution of Thioflavin T (ThT) (Sigma–Aldrich) in 0.5 × PBS and analyzed in a FluoroMax-3 spectrofluorometer (HORIBA Jobin Yvon, Bensheim, Germany) using an excitation wavelength of 450 nm and an emission wavelength of 482 nm. The integration time was set to 20 s with a slit width of 2 nm. Measured fluorescence intensities were set to zero for t = 0 and the asymptotic value of the fluorescence intensity of aggregated Aβ(1-40) in 0.5 × PBS was set to 100%.

## Confocal plaque imaging

For plaque staining, mice were injected intraperitoneally with Methoxy-X04 (1.7 mg/kg body weight). After 18 h, the mice were perfused intracardially with 4% PFA and the brains were post-fixed overnight[71]. The next day, coronal sections were prepared and Neu-roTrace deep red counterstaining was performed[72]. Slices were mounted in a confocal microscope (Olympus) and images were taken using a 4X air immersion objective and stitched together.

## Statistics and Reproducibility

For the comparison of two groups, we used two-tailed Wilcoxon signed-rank or rank sum tests. For the comparison of multiple groups, we used a Kruskal–Wallis test with Dunn-Sidac post-hoc comparison or a Kolmogorov-Smirnov-Test. The sigmoidal curve in Fig. S4B was created by non-linear regression, using SEM data weight, according to the Equation of Mass Law. Details on the exact statistical test used as well as the number of subjects and samples are included in the respective figure legends. N-numbers were equal to comparable two-photon in vivo imaging studies and explicitly experiments from our group and others investigating neuronal dysfunctions in AD[10,12,14,18,33]. No statistical method was used to predetermine sample size. When comparing the effects of the application of different proteins/peptides and controls, animals were randomized. Blinding was not possible, because, due to the neuronal hyperactivity in AD mice, their genotype would be immediately apparent to the experimenter.

## Reporting summary

Further information on research design is available in the Nature Portfolio Reporting Summary linked to this article.

## Data availability

The crystal structure of the Aβ-anticalin bound to Aβ 1-40[24] is available at the RCSB protein data bank under accession code 4MVL. The remaining data generated in this study are provided in the Supplementary Information/Source Data file. Raw data will be provided upon reasonable request to the corresponding authors. Source data are provided with this paper.

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

## Acknowledgements

We thank C. Karrer, F. Beyer, S. Achatz and G. Finck for technical support. We are grateful to L. Looger for providing SF-iGluSnFR constructs. This work was funded by the German Research Foundation (DFG grant no. 685472) to BZ and the Max Planck School of Cognition. BZ is a is an Albrecht-Struppler-Clinician Scientist Fellow, funded by the Federal Ministry of Education and Research (BMBF) and the Free State of Bavaria under the Excellence Strategy of the Federal Government and the Länder, as well as by the Technical University of Munich - Institute for Advanced Study. AK is a Hertie-Senior-Professor for Neuroscience. MAB is supported by the UK Dementia Research Institute, which receives its funding from DRI Ltd., funded by the Medical Research Council, Alzheimer's Society and Alzheimer Research UK and by a UKRI Future Leaders Fellowship (grant number: MR/S017003/1).

## Author contributions

B.Z., A.S and A.K. planned and oversaw all aspects of the study, B.Z., C.G., C.W., A.K-D and M.A.B performed and analyzed in vivo experiments, F.U. performed glutamate imaging experiments. M.M.K. performed in vitro hippocampal calcium imaging. L.N. and A.F. purified and characterized the Aβ-anticalin protein in vitro. B.Z., A.S. and A.K. wrote the manuscript with input and substantial revisions from all authors.

## Funding

## Competing interests

A.S. is founder and shareholder of Pieris Pharmaceuticals, Inc. Anticalin® is a registered trademark of Pieris Pharmaceuticals GmbH, Germany. All other authors declare no competing interests.
