## [Peer Review File · Nature Communications]

β -amyloid monomer scavenging by an anticalin protein prevents neuronal hyperactivity in mouse models of Alzheimer's DiseaseREVIEWER COMMENTS

Reviewer #1 (Remarks to the Author):

The manuscript by Zott and colleagues uses a multi-disciplinary approach to demonstrate the through preventing A β oligomerization, A β -anticalin can reduce hippocampal hyper-activity that has been suggested to contribute to cognitive deficits in Alzheimer's disease. Together, the results are compelling, both respect to the potential of anticalin therapies, as well as novel insight into the rate of A β oligomerization in the brain and its effects on neuronal activity. Addressing the points raised below will clarify the results presented, and increase the impact of the manuscript.

1. For the data presented in Figure 1, demonstrating the impact of AB-anticalin treatment on hippocampal neural activity in tissue from an AB-overexpression mouse model, I found myself curious as to whether the effects of anti-caline specific to the hyperactive cells? Specifically, the analysis shows a change in the proportion of hyperactive cells, but would be nice to actually quantify the impact on hyperactive cells themselves; e.g event rate went from >20/min to what in an identified "hyperactive" cell. This should be possible to extract from the 2-photon data. Related, how was 20 events/min picked as the definition of a hyperactive cell? Does this reflect a physiological state that impairs mnemonic function? From the data in Figure 1 and S2-3, it looks like there is a fairly large effect of anticalin treatment on cells in the 10-20 transients/min activity range. It also seems to have increased the number of hypo-active cells. How are do these results fit in with the larger story? Finally, what effect does the anticalin have on the activity in identified hyperactive cells in the WT mice? Addressing these issues clearly in the main text will help clarify the impact of the anticalin on cellular activity and its potential as a treatment.

2. What does anticalin treatment do to hyperactive cells in older mice? Seems relevant to any patient population that may receive such treatment, as identification of prodromal AD in the human population is rare, and given that in this strain it's older (6-8 mo) that show memory impairments. Indeed, as the authors' state, "Therefore, we conclude that application of the A β -anticalin is ineffective in preventing A β -dependent neuronal hyperactivation once toxic A β aggregates have formed." Would we then predict that enough oligomerization is still ongoing in later stage AD for such a treatment strategy to be efficacious?

3. An experiment showing that anticalin treatment in young AD model mice ameliorates cognitive impairments and/or neural pathology that occur in 6-8 month old mice would be impactful.

4. The apparently rapid rate of ongoing AB oligomerization was one of the more stunning implications of the data, though as much of the experimental evidence was generated using synthetic AB monomers, it begs the question of how rapidly oligomers form in vivo from endogenous sources, and how this compares to the rate oligomers are formed from synthetic AB monomers. Basically, are the rapid on and offset kinetics of the effects of anticalin on neural activity mesh with what is known about the rate of AB oligomerization in the brain? Related, are the

amounts/concentrations of synthetic monomers being used physiological? I.e., are they mimicking the in vivo environment, or are they adding so much the rate of oligomerization and such is now supra-physiological?

5. An experiment showing the impact of anticalin on glutamate transients in AD-model mice would more directly tie Figure 5 to Figure 1, and provide more mechanistic insight.

6. For both imaging experiments, further analysis of transient kinetics (AUC, decay rate) would also provide additional mechanistic insight. Related to this, in Figure 4, the impact of putative AB oligomers (“aged” AB solution, 4D), and AB oligomers in the presence of anticalin, is suggested to be the same. However, those traces suggest very different effects of the “aged” AB in the presence of the anticalin; transient frequency may be unaltered, but the amplitude/clearance seems rather different. These points are critical for understanding the potential effect of anticalin alone, and in the presence of AB monomers and oligomers.

7. How were doses/volumes of anticalin and AB monomers chosen? This ratio is key to understanding the potential as a pharmacotherapy.

Reviewer #2 (Remarks to the Author):

The authors report the impacts of an anticalin that binds Abeta monomer on neuronal hyperactivity.

The strengths of the paper are:

1. The experimental setup for measuring neuronal hyperactivity in vivo. The measurements are interesting and potentially important.
2. The findings in Figure 1 that show the anticalin can prevent neuronal hyperactivity. This is fascinating given that the anticalin seems to only bind Abeta monomer.

The weaknesses of the paper are:

1. The data in Figures 2-5 are not nearly as interesting as the data in Figure 1. After the exciting result in Figure 1, the other data simply demonstrates that the anticalin is inactive against Abeta dimers (Figure 2) and prevents Abeta aggregation if added to the monomer (Figure 4).
2. The lack of comparison of the anticalin to other Abeta binding agents (antibodies, antibody fragments) makes one wonder if any binding agent would do what the anticalin does. Do the authors believe that the anticalin binding specificity or activity is unique relative to other Abeta binding agents?

Reviewer #1 (Remarks to the Author):

The manuscript by Zott and colleagues uses a multi-disciplinary approach to demonstrate the through preventing A β oligomerization, A β -anticalin can reduce hippocampal hyper-activity that has been suggested to contribute to cognitive deficits in Alzheimer's disease. Together, the results are compelling, both respect to the potential of anticalin therapies, as well as novel insight into the rate of A β oligomerization in the brain and its effects on neuronal activity. Addressing the points raised below will clarify the results presented, and increase the impact of the manuscript.

We thank the reviewer for this positive feedback.

1. For the data presented in Figure 1, demonstrating the impact of AB-anticalin treatment on hippocampal neural activity in tissue from an AB-overexpression mouse model, I found myself curious as to whether the effects of anti-caline specific to the hyperactive cells?

Thank you for this comment. We have now included a thorough analysis of the effects of the A β -anticalin in a cell-by-cell fashion. (Fig. S4 A, B). It is important to note that the A β -anticalin treatment not only reduced the number of hyperactive cells, but induced, in treated APP23xPS45 mice, an activity pattern which is not discernable from that of WT mice (Fig. 1 F). We have put more emphasis on this point in the revised manuscript. The reason we focused on the number of hyperactive cells in the manuscript is that they represent the most striking difference between the two genotypes.

Specifically, the analysis shows a change in the proportion of hyperactive cells, but would be nice to actually quantify the impact on hyperactive cells themselves; e.g event rate went from >20/min to what in an identified "hyperactive" cell. This should be possible to extract from the 2-photon data.

We have now included the analysis of the activity levels of every individual hyperactive cell in wild type and APP23xPS45 mice in Fig. S3 C and D.

Related, how was 20 events/min picked as the definition of a hyperactive cell? Does this reflect a physiological state that impairs mnemonic function?

The definition of a hyperactive neuron was in accordance with previous publications (Busche et al 2012, Busche et al 2008, Zott et al 2019) and is based on the distribution of activity levels in wildtype mice. In principle, hyperactive neurons have a higher frequency than 97-98% of the entire population of wild type neurons.

From the data in Figure 1 and S2-3, it looks like there is a fairly large effect of anticalin treatment on cells in the 10-20 transients/min activity range. It also seems to have increased the number of hypo-active cells. How are do these results fit in with the larger story?

Thank you for this comment. We feel that the results are perfectly in line with the larger story. In our A β -application experiments in wild-type mice, we have previously observed that A β increases the number of Ca²⁺-transients in all neurons to a degree which is dependent on their individual level of baseline activity (Zott et al 2019). In consequence, in an AD mouse even neurons with "normal" activity levels actually are more active than they would be in the absence of A β , except for those with very low baseline activity. Thus, a reduction in neuronal activity across all neurons, which is more pronounced in the most active neuronal population, is actually expected if A β is neutralized. Also, the number of silent cells should increase to a percentage similar to that observed in wild type mice. In fact, this is exactly what we observe (Fig. S4 A, B). Moreover, the distribution of neuronal activity levels was not significantly different between treated APP23xPS45 and wild type mice (Fig. 1F).

Finally, what effect does the anticalin have on the activity in identified hyperactive cells in the WT mice?

We have now included the analysis of the activity levels of every hyperactive cell in wild type and APP23 x PS45 mice in Fig. S3 C and D. The A β -anticalin did not systematically change the activity levels in the very few hyperactive cells in WT mice.

Addressing these issues clearly in the main text will help clarify the impact of the anticalin on cellular activity and its potential as a treatment.

2. What does anticalin treatment do to hyperactive cells in older mice? Seems relevant to any patient population that may receive such treatment, as identification of prodromal AD in the human population is rare, and given that in this strain it's older (6-8 mo) that show memory impairments. Indeed, as the authors' state, "Therefore, we conclude that application of the A β -anticalin is ineffective in preventing A β -dependent neuronal hyperactivation once toxic A β aggregates have formed." Would we then predict that enough oligomerization is still ongoing in later stage AD for such a treatment strategy to be efficacious?

Based on our observations, we hypothesized that removing A β monomers by anticalins will only be efficacious at early disease stages, when oligomers are still mainly formed from (nascent) monomers and not released from larger aggregates, i.e., plaques. To test this, we have now included an experiment in which we applied the A β -anticalin in older AD mice, which did not reduce neuronal hyperactivity. We conclude that the A β -anticalin treatment will not be sufficient in late disease stages, but it may be a suitable add-on for treatment strategies aiming at the dissolution or removal of larger aggregates by preventing the formation of new oligomers.

3. An experiment showing that anticalin treatment in young AD model mice ameliorates cognitive impairments and/or neural pathology that occur in 6-8 month old mice would be impactful.

We agree. However, this is beyond the scope of this article, which demonstrates the efficacy of acute A β -anticalin treatment on neuronal activity. Moreover, we do not expect the A β -anticalin in its present molecular format to efficiently cross the brain-blood-barrier, which makes longitudinal experiments hard to realize at this stage. We are, however, developing a blood-brain-barrier permeant version of the A β -anticalin, which should make long-term experiments feasible in the future.

4. The apparently rapid rate of ongoing AB oligomerization was one of the more stunning implications of the data, though as much of the experimental evidence was generated using synthetic AB monomers, it begs the question of how rapidly oligomers form in vivo from endogenous sources, and how this compares to the rate oligomers are formed from synthetic AB monomers. Basically, are the rapid on and offset kinetics of the effects of anticalin on neural activity mesh with what is known about the rate of AB oligomerization in the brain?

This is an excellent point. To our knowledge, the rate of oligomerization in vivo is unknown because reliable biomarkers do not exist. However, the total turnover rate of soluble A β can be determined after gamma-secretase inhibition. The half-life of A β is around one hour after oral gavage of different gamma-secretase inhibitors (Abramowski et al 2008, Barten et al 2005). In consequence, it is reasonable to assume that after direct application of A β onto the respective neurons, clearance is even faster. To test this, we now performed an additional experiment in which we applied a gamma secretase inhibitor the same way as the A β -anticalin in APP23xPS45 mice – leading to very similar effects as seen for the A β -anticalin, i.e. a rapid reduction in hyperactivity which washed out after 10-

15 minutes (Fig. S6 and S10). This data independently confirms that local A β turnover (including oligomerization) is very rapid in vivo.

In contrast, γ -secretase inhibition, like A β -anticalin application, also had no effect in older mice. This supports our hypothesis that in old plaque-bearing AD mice scavenging of A β monomers or preventing their release alone is not sufficient to restore normal neuronal activity patterns. This might be because the formation of toxic oligomers at this stage is mainly dependent on the release from aggregates or plaques, rather than due to de novo formation.

Related, are the amounts/concentrations of synthetic monomers being used physiological? I.e., are they mimicking the in vivo environment, or are they adding so much the rate of oligomerization and such is now supra-physiological?

We used different concentrations of A β , ranging from 1 to 100 μ M monomer equivalent. This is comparable to what we and other groups used previously (Busche et al 2015, Shankar et al 2008, Zott et al 2019). The concentration of A β in the brain in vivo is lower, but the overall turnover rate of A β is still very rapid (see above), possibly because the A β in the brain is unevenly distributed, showing 'hotspots' with very high concentrations. It is important to note that, in our hands, the A β -anticalin prevented the formation of oligomers and aggregates in various experiments, independent of the A β concentration applied (Fig. 4 B, G, S7).

5. An experiment showing the impact of anticalin on glutamate transients in AD-model mice would more directly tie Figure 5 to Figure 1, and provide more mechanistic insight.

Thank you for this suggestion. We have performed new experiments using an updated version of the iGlu SnFr-sensor (Fig. 6). We are pleased that our data not only demonstrate glutamate accumulation in AD mice compared to wild-type animals but also the restoration of the glutamate homeostasis by the A β -anticalin. These findings are valuable as they independently confirm the results in Fig. 1 and provide further insights into the mechanism of neuronal hyperactivity in AD.

6. For both imaging experiments, further analysis of transient kinetics (AUC, decay rate) would also provide additional mechanistic insight. Related to this, in Figure 4, the impact of putative AB oligomers ("aged" AB solution, 4D), and AB oligomers in the presence of anticalin, is suggested to be the same. However, those traces suggest very different effects of the "aged" AB in the presence of the anticalin; transient frequency may be unaltered, but the amplitude/clearance seems rather different. These points are critical for understanding the potential effect of anticalin alone, and in the presence of AB monomers and oligomers.

Thank you for this comment. We have now added an analysis of the AUC for these experiments in Fig. S8 which confirm our analysis of the number of Ca²⁺-transients. The apparent difference between Fig. 4E and I may be explained by a slight variation between the in vivo experiments (note also the differences under baseline conditions).

More importantly, we have now included an experiment, in which we applied the A β -anticalin in older plaque-bearing AD mice (Fig. S6), which showed no effect on neuronal activity. This independently confirms the notion that the A β -anticalin is ineffective against aggregated A β .

7. How were doses/volumes of anticalin and AB monomers chosen? This ratio is key to understanding the potential as a pharmacotherapy.

In the application experiments, we used stoichiometric ratios, i.e., one molecule A β -anticalin for one molecule A β . Previous in vitro data suggest that substoichiometric ratios of the A β -anticalin already significantly suppress aggregate formation (Rauth et al 2016).

Reviewer #2 (Remarks to the Author):

The authors report the impacts of an anticalin that binds Abeta monomer on neuronal hyperactivity.

The strengths of the paper are:

1. The experimental setup for measuring neuronal hyperactivity in vivo. The measurements are interesting and potentially important.
2. The findings in Figure 1 that show the anticalin can prevent neuronal hyperactivity. This is fascinating given that the anticalin seems to only bind Abeta monomer.

We thank the reviewer for this assessment.

The weaknesses of the paper are:

1. The data in Figures 2-5 are not nearly as interesting as the data in Figure 1. After the exciting result in Figure 1, the other data simply demonstrates that the anticalin is inactive against Abeta dimers (Figure 2) and prevents Abeta aggregation if added to the monomer (Figure 4).

We respectfully disagree. Figures 2-4 provide important mechanistic insights into the action of the A β -anticalin as discussed in our response to Reviewer #1 above. In the revised manuscript, we now have added additional experimental data further confirming our main hypotheses as explained above. Additionally, the new data shown in Fig. 6 related to the restoration of glutamate homeostasis provide further insights into the mechanism of neuronal hyperactivity in AD and independently confirm the findings from Fig.1

2. The lack of comparison of the anticalin to other Abeta binding agents (antibodies, antibody fragments) makes one wonder if any binding agent would do what the anticalin does. Do the authors believe that the anticalin binding specificity or activity is unique relative to other Abeta binding agents?

We thank the reviewer for this comment. We now demonstrate that not only scavenging A β monomers but also preventing their release (Fig. S6) reduces neuronal hyperactivity in young APP23xPS45 mice.

Although, in principle, any agent that effectively removes A β monomers should reduce neuronal hyperactivity, the monomer-scavenging ability of the A β -anticalin may be unique. In this context, the epitope that is recognized (by the anticalin or other binding proteins) within the A β peptides, which represent pretty long antigenic sequences, may play a role, as previously discussed (Rauth et al., 2016). Furthermore, in comparison to the antibody solanezumab, which binds a similar central epitope of the A β peptide, the A β -anticalin binds the monomer in a distinct zig-zag confirmation, which may represent an intermediary step on the path towards oligomerization (Eichinger et al 2022). The similarities and differences between the A β -anticalin and antibodies have been discussed in detail (Eichinger et al 2022). There, it was also experimentally demonstrated that the A β -anticalin shows much less cross-reactivity with endogenous (plasma) proteins than solanezumab, for example. Some of these aspects have been mentioned at the end of the discussion.

- Abramowski D, Wiederhold KH, Furrer U, Jatton AL, Neuenschwander A, et al. 2008. Dynamics of A β turnover and deposition in different beta-amyloid precursor protein transgenic mouse models following gamma-secretase inhibition. *J Pharmacol Exp Ther* 327: 411-24
- Barten DM, Guss VL, Corsa JA, Loo A, Hansel SB, et al. 2005. Dynamics of β -Amyloid Reductions in Brain, Cerebrospinal Fluid, and Plasma of β -Amyloid Precursor Protein Transgenic Mice Treated with a γ -Secretase Inhibitor. *Journal of Pharmacology and Experimental Therapeutics* 312: 635-43
- Busche MA, Chen X, Henning HA, Reichwald J, Staufenbiel M, et al. 2012. Critical role of soluble amyloid-beta for early hippocampal hyperactivity in a mouse model of Alzheimer's disease. *Proc Natl Acad Sci U S A* 109: 8740-5
- Busche MA, Eichhoff G, Adelsberger H, Abramowski D, Wiederhold KH, et al. 2008. Clusters of hyperactive neurons near amyloid plaques in a mouse model of Alzheimer's disease. *Science* 321: 1686-9
- Busche MA, Kekus M, Adelsberger H, Noda T, Forstl H, et al. 2015. Rescue of long-range circuit dysfunction in Alzheimer's disease models. *Nat Neurosci* 18: 1623-30
- Eichinger A, Rauth S, Hinz D, Feuerbach A, Skerra A. 2022. Structural basis of Alzheimer beta-amyloid peptide recognition by engineered lipocalin proteins with aggregation-blocking activity. *Biological chemistry* 403: 557-71
- Rauth S, Hinz D, Borger M, Uhrig M, Mayhaus M, et al. 2016. High-affinity Anticalins with aggregation-blocking activity directed against the Alzheimer beta-amyloid peptide. *Biochem J* 473: 1563-78
- Shankar GM, Li S, Mehta TH, Garcia-Munoz A, Shepardson NE, et al. 2008. Amyloid-beta protein dimers isolated directly from Alzheimer's brains impair synaptic plasticity and memory. *Nat Med* 14: 837-42
- Zott B, Simon MM, Hong W, Unger F, Chen-Engerer H-J, et al. 2019. A vicious cycle of β amyloid-dependent neuronal hyperactivation. *Science* 365: 559-65

REVIEWER COMMENTS

Reviewer #1 (Remarks to the Author):

The authors were very responsive to the previous round of review, and added interesting new results. My comments below are meant to highlight issues with the current presentation of the data that make it difficult for the impact of the data to come through.

Lines 86-107; written in a way that is difficult to follow. If I understand it correctly, AB anticalin treatment reduced activity in all cells of APP mice, though the effect was strongest for those cells with the highest basal activity rate (I would state in the text how this is defined). AB anticalin has no effect on activity of cells in WT mice.

The reader is then presented with the lack of effect of the AB anticalin on synthetic dimer-induced hyperactivity, followed by evidence of the apparent affinity of the AB anticalin for monomers; synthetic versions of which have no effect on hyperactivity and then followed by the effect of the secretase inhibitor (presumably scavenging monomers) on hyperactivity.

These data are then cast in the light of the apparent importance of blocking oligomerization. The paper concludes with the glutamate data, which is interesting but would be more impactful if framed more explicitly in the context of explaining the hyperactivity results that precede it; e.g. if these slices are also 2-3 mo old mice as those in the Ca²⁺ imaging from figure one.

As opposed to breaking the paper up into sections describing the individual experiments and results, I would suggest re-organizing along the lines of deducing what these AB anticalins are doing to monomer/dimer/oligomer effects. The authors do discuss this issue in the current version of the manuscript, my point is that the way the argument is presented requires the reader to jump back and forth between figures and sections of the results to understand the bigger picture.

Minor note, it would be nice to acknowledge that increase Glu release is one, of multiple, mechanisms that may contribute to increased Ca²⁺ transients.

Reviewer #3 (Remarks to the Author):

This is a revision of a manuscript in which the authors report reduction of neuronal hyperactivity in transgenic mice with direct hippocampal application of an anticalin that binds Abeta monomers.

The methodology described in the manuscript seems overall quite sound and the data tell a consistent story throughout.

The authors apply an impressive number of sophisticated techniques to answer their experimental

questions: two photon microscopy, live cell calcium imaging, viral expression of a fluorescent glutamate sensor, etc. Ultimately, their results make a convincing case that by binding Abeta monomer, the anticalin is able to prevent formation of toxic oligomers, reduce accumulation of extracellular glutamate, and decrease neuronal hyperactivity - at least in a transgenic, abeta driven model of AD.

At the same time, the major weakness of the manuscript is that the findings are presented as though they have translational significance or near term potential for therapeutic application. While perhaps a useful experimental tool, there is little evidence presented to indicate that the abeta anticalin is a viable translational therapeutic. Pharmacokinetics are never characterized, and while experiments are performed "in vivo", the anticalin is never tested with any real pharmacologic rigor. Not only is it directly applied to neuronal tissue via injection pipette, but a single, exceedingly high concentration (10uM) is used. If the authors' intention were truly to develop the protein as a pre-clinical therapeutic, it seems likely that they would want to assess its dose response or at least try to estimate the parenchymal concentration needed to produce therapeutic effects. While they may be developing a "BBB-permeant version" of the anticalin, as they mention in their rebuttal, they cannot be thinking that they will achieve sustained parenchymal concentrations in the micromolar range. Even the most promising contemporary pharmacologic strategies for biologic delivery across the BBB (e.g., Denali's "transport vehicle") typically aim for single or low double-digit nanomolar concentrations.

Likewise, the response regarding other Abeta binding agents is unconvincing. In both the rebuttal and discussion, the authors mention a unique "zig-zag" binding conformation and describe a number of potential advantages of anticalins over IgGs, but there are no data presented to support these assertions. The unfortunate truth is that by insisting that their results suggest a promising strategy for preventive treatment of AD, they detract from what is otherwise a sound scientific study with intriguing mechanistic findings.

REVIEWER COMMENTS

Reviewer #1 (Remarks to the Author):

The authors were very responsive to the previous round of review, and added interesting new results. My comments below are meant to highlight issues with the current presentation of the data that make it difficult for the impact of the data to come through.

We thank reviewer #1 for this comment. See the detailed responses below.

Lines 86-107; written in a way that is difficult to follow. If I understand it correctly, AB anticalin treatment reduced activity in all cells of APP mice, though the effect was strongest for those cells with the highest basal activity rate (I would state in the text how this is defined). AB anticalin has no effect on activity of cells in WT mice.

Thank you for this comment. Yes, that is correct. We have now rephrased the text to make this point clearer and included the definition of hyperactivity.

The reader is then presented with the lack of effect of the AB anticalin on synthetic dimer-induced hyperactivity, followed by evidence of the apparent affinity of the AB anticalin for monomers; synthetic versions of which have no effect on hyperactivity and then followed by the effect of the secretase inhibitor (presumably scavenging monomers) on hyperactivity.

These data are then cast in the light of the apparent importance of blocking oligomerization. The paper concludes with the glutamate data, which is interesting but would be more impactful if framed more explicitly in the context of explaining the hyperactivity results that precede it; e.g. if these slices are also 2-3 mo old mice as those in the Ca²⁺ imaging from figure one.

Thank you for this comment. We have now restructured the manuscript by moving former Fig. 5 up (now Fig. 2), as suggested. We agree that this makes the point that the A β -anticalin reduces hyperactivity and restores glutamate homeostasis much clearer.

As opposed to breaking the paper up into sections describing the individual experiments and results, I would suggest re-organizing along the lines of deducing what these AB anticalins are doing to monomer/dimer/oligomer effects. The authors do discuss this issue in the current version of the manuscript, my point is that the way the argument is presented requires the reader to jump back and forth between figures and sections of the results to understand the bigger picture.

We have now restructured the results section according to this suggestions and broken it up into three distinct parts. We feel that the main story is much easier to follow now.

Minor note, it would be nice to acknowledge that increase Glu release is one, of multiple, mechanisms that may contribute to increased Ca²⁺ transients.

In the introduction, we now acknowledge other explanations for hyperactivity in mice.

Reviewer #3 (Remarks to the Author):

This is a revision of a manuscript in which the authors report reduction of neuronal hyperactivity in transgenic mice with direct hippocampal application of an anticalin that binds Abeta monomers.

The methodology described in the manuscript seems overall quite sound and the data tell a consistent story throughout.

The authors apply an impressive number of sophisticated techniques to answer their experimental questions: two photon microscopy, live cell calcium imaging, viral expression of a fluorescent glutamate sensor, etc. Ultimately, their results make a convincing case that by binding Abeta monomer, the anticalin is able to prevent formation of toxic oligomers, reduce accumulation of extracellular glutamate, and decrease neuronal hyperactivity - at least in a transgenic, abeta driven model of AD.

Thank you for this assessment.

At the same time, the major weakness of the manuscript is that the findings are presented as though they have translational significance or near term potential for therapeutic application. While perhaps a useful experimental tool, there is little evidence presented to indicate that the abeta anticalin is a viable translational therapeutic.

We thank the reviewer for this comment. Our main intention was to demonstrate that A β monomer scavenging can be effective in reducing neuronal and synaptic dysfunctions by a mechanism involving the prevention of oligomer formation. We did not intend to give the impression that the A β -anticalin has “translational significance or near term potential for therapeutic intervention”. Still, our findings should encourage further investigation into the possibility of an anticalin-based therapeutic intervention.

To make these points clearer and to avoid misunderstandings, we have now rewritten the abstract, introduction and discussion. See also our detailed responses below.

Pharmacokinetics are never characterized, and while experiments are performed "in vivo", the anticalin is never tested with any real pharmacologic rigor. Not only is it directly applied to neuronal tissue via injection pipette, but a single, exceedingly high concentration (10uM) is used. the authors' intention were truly to develop the protein as a pre-clinical therapeutic, it seems likely that they would want to assess its dose response or at least try to estimate the parenchymal concentration needed to produce therapeutic effects

As suggested by reviewer #3, we have now tested the in vivo application of the A β -anticalin at different pipette concentrations (Fig. S4). We have modeled a dose-response curve and found the IC₅₀ in the pipette tip to be 75 nM.

While they may be developing a "BBB-permeant version" of the anticalin, as they mention in their rebuttal, they cannot be thinking that they will achieve sustained parenchymal concentrations in the micromolar range. Even the most promising contemporary pharmacologic strategies for biologic delivery across the BBB (e.g., Denali's "transport vehicle") typically aim for single or low double-digit nanomolar concentrations.

It is important to remember that the concentration in the pipette is higher than what can be achieved by pressure ejection of the drug into the surrounding tissue. In fact, the parenchymal

concentration of a drug applied from a pipette rapidly decreases with increasing distance to the tip (Kirkpatrick et al 2014, Kirkpatrick & Wightman 2016). In consequence, based on the IC_{50} of 75 nM in the pipette (see above) and the distance of the investigated neurons and synapses from the pipette tip (ranging from tens to a few hundred μm , compare e.g. Fig. 1C and 2B), we expect the parenchymal concentration needed to produce therapeutic effects to be close to the “single or low double-digit nanomolar concentrations” mentioned by reviewer 3.

Likewise, the response regarding other Abeta binding agents is unconvincing. In both the rebuttal and discussion, the authors mention a unique "zig-zag" binding conformation and describe a number of potential advantages of anticalins over IgGs, but there are no data presented to support these assertions.

We have now included a comparison of the $A\beta$ -anticalin with a Solanezumab biosimilar antibody (Fig. S6). We have chosen this antibody because it has the same $A\beta$ binding site as the anticalin, and, consequently, a high affinity for $A\beta$ monomers. At the same pipette concentration, Solanezumab also reduced neuronal activity levels in AD mice, but to a lesser extent than the $A\beta$ -anticalin. However, we agree that it is not entirely clear what causes these differences in efficacy. Consequently, we have removed any discussion of the possible reasons, including the distinct binding conformation of the $A\beta$ -monomer, from the manuscript.

The unfortunate truth is that by insisting that their results suggest a promising strategy for the preventive treatment of AD, they detract from what is otherwise a sound scientific study with intriguing mechanistic findings.

We thank the reviewer for the final assessment. In response to all comments, we have now rewritten the last paragraph of the discussion and rephrased the introduction and the abstract to highlight the mechanistic findings and the methodology rather than possible therapeutic implications. Still, although it is still too early to say whether the $A\beta$ -anticalin will ever enter or succeed in clinical trials, we feel that our findings, including the new experiments in Fig. S4 and S6, should at least encourage further investigation into the possibility of an $A\beta$ -anticalin based therapeutic strategy.

References:

- Kirkpatrick DC, Edwards MA, Flowers PA, Wightman RM. 2014. Characterization of Solute Distribution Following Iontophoresis from a Micropipet. *Analytical Chemistry* 86: 9909-16
- Kirkpatrick DC, Wightman RM. 2016. Evaluation of Drug Concentrations Delivered by Microiontophoresis. *Anal Chem* 88: 6492-9

REVIEWERS' COMMENTS

Reviewer #1 (Remarks to the Author):

I appreciate the authors' responsiveness to the multiple rounds of critique. I believe the manuscript is much stronger, and I have no further substantive comments.

Reviewer #3 (Remarks to the Author):

The authors have significantly improved their manuscript in response to the prior review. In particular, they now provide evidence of dose response to anticalin with estimation of an IC50.

While 75nM is still a pretty high concentration for brain parenchyma, its far better than 10uM and the authors' argument about the relationship between the concentration in the pipette tip and the concentration in the surrounding tissue seems valid and well supported by citations.

Most importantly, they now clarify that this is primarily a mechanistic study which of course could have therapeutic implications. It is my opinion that the paper is now appropriate for publication.